# Fast $L_2$ Calibration for Inexact Highway Traffic Flow Systems

**Jingru Huang** [1], **Yan Wang** [1,*] and **Mei Han** [2]

1   School of Statistics and Data Science, Faculty of Science, Beijing University of Technology,
    Beijing 100124, China
2   College of Economics and Management, Nanjing University of Aeronautics and Astronautics,
    Nanjing 211106, China
*   Correspondence: yanwang@bjut.edu.cn

**Abstract:** Transportation systems need more accurate predictions to further optimize traffic network design with the development and application of autonomous driving technology. In this article, we focus on highway traffic flow systems that are often simulated by the modified Greenshields model. However, this model can not perfectly match the true traffic flow due to its underlying simplifications and assumptions, implying that it is inexact. Specifically, some parameters affect the simulation accuracy of the modified Greenshields model, while tuning these parameters to improve the model's accuracy is called model calibration. The parameters obtained using the $L_2$ calibration have the advantages of high accuracy and small variance for an inexact model. However, the method is calculation intensive, requiring optimization of the integral loss function. Since traffic flow data are often massive, this paper proposes a fast $L_2$ calibration framework to calibrate the modified Greenshields model. Specifically, the suggested method selects a sub-design containing more information on the calibration parameters, and then the empirical loss function obtained from the optimal sub-design is utilized to approximate the integral loss function. A case study highlights that the proposed method preserves the advantages of $L_2$ calibration and significantly reduces the running time.

**Keywords:** traffic flow system; modified greenshields model; sequential sub-design; $L_2$ calibration; uncertainty quantification

## 1. Introduction

To keep up with rapidly growing travel demands, urban traffic management systems are required to be continuously updated and innovated. Among them, connected and automated vehicles (CAV) are considered to be new technologies with great promise, as well as the future direction of the global automotive industry. CAV needs more accurate traffic dynamics at the network level to secure transport infrastructure and to prevent traffic congestion [1]. Thus, it is mandatory to analyze more extensively the characteristics of spatio-temporal travel patterns for traffic flow analysis at the network level. Given that the dynamic traffic assignment system (DTA) is often utilized to simulate real traffic flows [2,3], the DTA's dual-regime modified Greenshields traffic flow model can be employed to simulate highway traffic based on previous experience [4], called computer model or determetic simulator in computer experiments. The model can be expressed as a set of segmentation functions:

$$v_l = \begin{cases} u_f, & 0 < k_l < k_{bp}, \\ v_0 + (v_f - v_0)(1 - \dfrac{k_l}{k_{jam}})^{\alpha}, & k_{bp} < k_l < k_{jam}, \end{cases} \tag{1}$$

where $v_l$ is the speed on link $l$ on which we are focusing, $k_l$ is the density on link $l$ indirectly determined by the flow rate to speed ratio, i.e., $k_l = f_l/v_l$, and $f_l$ denotes the

total carriageway flow on link $l$. The density is the input variable of interest, referred to as *design* in the computer experiments. Moreover, $u_f$, $v_0$, and $v_f$ are the free-flow speed, minimum speed, and intercept speed on link $l$, respectively, $k_{bp}$ and $k_{jam}$ are the breakpoint density and the jam density on link $l$, and $\alpha$ is a shape parameter. Figure 1 illustrates the modified Greenshields model.

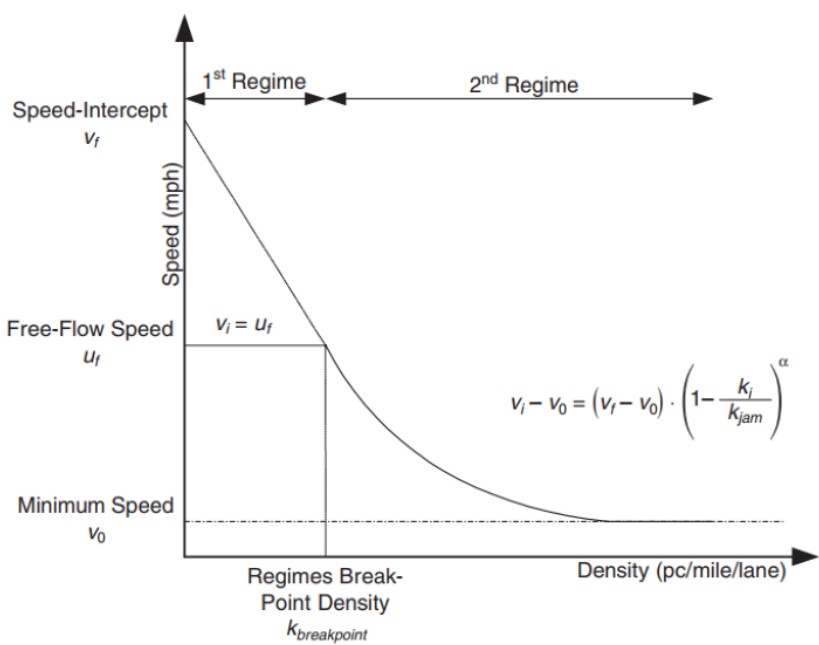

**Figure 1.** The dual-regime modified Greenshields traffic flow model of DTA.

This is because the vehicles' speeds are different for different times for when the highway is not jammed, i.e., $v_0$ varies. Additionally, the speed changes due to the weather and terrain, and there are some ideal assumptions and simplifications in the dual-regime modified Greenshields model compared with the real traffic flow system [5–7]. Namely, the modified Greenshields model is *inexact*.

Let $\boldsymbol{\theta} = (k_{bp}, u_f, v_f, \alpha, v_0, k_{jam})^T$ be unknown or unobservable in the traffic flow system, with $\boldsymbol{\theta}$ typically affecting the reliability and credibility of the simulators' outputs. The process of adjusting these parameters utilizing real traffic flow data is called the *calibration* of computer models, and these parameters are the *calibration parameters* [8]. For further details on computer model calibration, the reader is referred to [9,10]. The literature has attempted several times to obtain accurate and rapid estimates of the calibration parameters, with many efforts focusing on obtaining a consistent estimator for the calibration parameters, which are then applied to the traffic flow framework. Current methods are the KO calibration [9], the least square calibration (LS) [11–13], the weighted least squares method (WLS) [14], and the optimization-based model calibration [3,15].

In [16,17], the authors defined the "true values" of the calibration parameters by minimizing the distance between the computer model and the physical system. A follow-up work, [16], proposed the $L_2$ calibration method that is one of the most widely used in practice [18,19]. This method is proven to have good statistical properties, including high accuracy and small variance, posing it an appealing calibration solution requiring small sample sizes. A brief review of this method is presented in Section 2. However, due to the large-scale traffic flow framework in practice, the $L_2$ calibration procedure involves complex integration operations that are calculation intensive. Therefore, to apply the $L_2$ calibration framework to the traffic flow model, we develop a fast $L_2$ calibration framework to obtain the estimates and variances of the calibration parameters from real data. Our contributions can be summarized as follows:

- We propose a fast $L_2$ calibration framework to estimate the calibration parameters. The suggested method finds the optimal sub-design containing more information about the calibration parameters. Then, the empirical $L_2$ loss function constructed from the sub-design is used to approximate the integral $L_2$ loss function.
- We develop the algorithm to generate sequential optimal sub-designs based on the proposed criterion.
- A bootstrap method is adopted to quantify the uncertainty of the calibration parameters.

The remainder of this article is organized as follows. Section 2 briefly reviews the $L_2$ calibration approach proposed in [16]. Then, this section introduces the proposed fast $L_2$ framework utilizing a sub-design criterion and empirical loss function. The algorithm generating the sequential optimal sub-design and the bootstrap method to quantify the uncertainty are also provided. Section 3 applies the proposed method to the traffic flow model of the M25 motorway in London, and finally, Section 4 concludes this work and discusses the findings.

## 2. Optimal Sub-Design for the $L_2$ Calibration

In Section 2.1, we review the $L_2$ calibration, and in Section 2.2, we explain how to develop the experimental sub-designs and how to quickly estimate the calibration parameters of the dual-regime modified Greenshields model. The algorithm for generating sequentially optimal designs is provided in Section 2.3, and finally, Section 2.4 suggests a bootstrap method to quantify the uncertainty.

### 2.1. A Review of the $L_2$ Calibration

We assume that the record traffic flow data $\mathbf{V} = (v_1^t, \cdots, v_n^t)^T$ are conducted at the design points $\mathbf{K} = (k_1, \cdots, k_n)^T$, where $k_i \in \mathbb{K}$ is a design value of the density. Suppose that $\zeta(\cdot)$ is the real traffic flow system, which is unknown. Since the measurement error always exists, the data can be presented as:

$$v_i^t = \zeta(k_i) + \varepsilon_i, \ i = 1, \cdots, n, \tag{2}$$

where $\varepsilon_i'$s are independent and identically distributed random variables with zero mean and finite variance $\tau^2 > 0$.

Since we are concerned with the traffic flows on only one link, the subscript $l$ from the notations in the dual-regime modified Greenshields model (1) is deleted thereafter. Let $v(k, \boldsymbol{\theta})$ be the output of the dual-regime modified Greenshields model, where $k \in \mathbb{K}$ indicates the density; $\boldsymbol{\theta} = \{k_{bp}, u_f, v_f, \alpha, v_0, k_{jam}\} \in \Theta$ is a set of the calibration parameters. The calibration process aims to find the estimates of $\boldsymbol{\theta}$ so that the modified Greenshields model outputs are as close as possible to the recorded data. Since the modified Greenshields model is inexact, there is a "distance" between $\zeta(\cdot)$ and $v(\cdot, \cdot)$, called the *discrepancy function*. Therefore, the relationship between the simulation model and the discrepancy function can be established as follows:

$$\zeta(\cdot) = v(\cdot, \boldsymbol{\theta}^\star) + \delta(\cdot), \tag{3}$$

where $\delta(\cdot)$ is the model discrepancy, which is an unknown function. In this article, we consider the observation error and model uncertainty during the calibration and prediction process of the traffic flow simulation model. Moreover, $\boldsymbol{\theta}^\star \in \Theta$ is the "true value" or the optimal calibration parameter, defined as [17]:

$$\boldsymbol{\theta}^\star = \underset{\boldsymbol{\theta} \in \Theta}{\operatorname{argmin}} \int_{\mathbb{K}} (\zeta(k) - v(k, \boldsymbol{\theta}))^2 dk. \tag{4}$$

This loss function is named as the $L_2$ *loss* function. In [16], the authors proposed the $L_2$ calibration method, where the definition of the calibration parameter estimation is:

$$\hat{\boldsymbol{\theta}}^{L_2} = \underset{\boldsymbol{\theta} \in \Theta}{\operatorname{argmin}} \int_{\mathbb{K}} (\hat{\zeta}(k) - v(k, \boldsymbol{\theta}))^2 dk, \tag{5}$$

where the optimization function is denoted as the $L_{L_2}$ loss function and $\hat{\zeta}(\cdot)$ is a nonparametric estimator of the traffic flow system estimated from the record data. The frequently used estimators include the Gaussian process models [20,21], kernel ridge regression [16], and smooth spline regression [22].

### 2.2. Optimal Sub-Design Criterion

$L_2$ calibration requires optimization of the functions containing integral operations. When the gradient descent algorithm is used, we must calculate the gradient and the Hessian matrix of the $L_2$ loss function to estimate the calibration parameters. These calculations involve complex integration operations because the integration needs to be recomputed at each update step. Thus, (4) poses a very challenging optimization process, especially for large-scale network systems such as traffic flow. To overcome this concern, the MCMC method approximates the $L_{L_2}$ loss integration. Indeed, a discrete set is generated from the design region $\mathbb{K}$, denoted $\{\xi_1, \cdots, \xi_M\}$, and the approximate loss is obtained as follows:

$$\hat{L}_{L_2}(\boldsymbol{\theta}) = \frac{1}{M} \sum_{i=1}^{M} (\hat{\zeta}(\xi_i) - v(\xi_i, \boldsymbol{\theta}))^2. \tag{6}$$

The minimum value of $\hat{L}_{L_2}$ within $\Theta$ is noted as $\hat{\boldsymbol{\theta}}$, which can be made arbitrarily near to the minimum of the $L_{L_2}$. However, the value of the approximate loss function imposes a significant computational burden, since $M$ is a large number. Therefore, we aim to design efficient samples to adjust the calibration parameters accurately under a certain criterion. In other words, we need to search for the optimal sub-design in the design region so that $\hat{\boldsymbol{\theta}}$ is as close as possible to $\boldsymbol{\theta}^\star$.

Our motivation is derived from the truncated least squares (LTS) [23] concept, which uses a portion of the selected samples by sorting the absolute values of the residuals. The proposed approach employs a similar idea to select the design with a large discrepancy over region $\mathbb{K}$, affording more robust calibration parameter estimates. Thus, the optimal sub-design can be obtained by maximizing the discrepancy function for a given value of $\boldsymbol{\theta}$. Suppose that we have a design of $N$ runs to estimate the calibration parameters efficiently. The non-parametric approximation of the highway traffic flow system is estimated employing the record flow data, which affords considering the system as a known model. Let the optimal sequential design be $\boldsymbol{k}^\star = \{k_1^\star, \cdots, k_N^\star\} \in \mathbb{K}^N$, then the first optimality criterion is:

$$\boldsymbol{k}^\star = \underset{\boldsymbol{k} \in \mathbb{K}^N}{\operatorname{argmax}} \|\hat{\zeta}(\boldsymbol{k}) - v(\boldsymbol{k}, \boldsymbol{\theta}^\star)\|, \tag{7}$$

where $\|\cdot\|$ denotes the Euclidean distance. Additionally, the optimal sub-design is expected to contain more information of the calibration parameters, which is a concept that is commonly used during the experimental design [20,24,25]. Specifically, this involves placing as many points as possible where the information of $\boldsymbol{\theta}$ is large affords robust and accurate estimates. Furthermore, based on the information maximization criterion, [24] suggested that the Fisher information matrix (FIM) of $\boldsymbol{\theta}$ is obtained by:

$$\mathbf{I}(\boldsymbol{k}, \boldsymbol{\theta}) = \sum_{i=1}^{N} \nabla v(k_i, \boldsymbol{\theta}) \nabla v(k_i, \boldsymbol{\theta})^T, \tag{8}$$

where $\nabla v(k_i, \boldsymbol{\theta}) = (\frac{\partial v(k_i, \boldsymbol{\theta})}{\partial k_{bp}}, \cdots, \frac{\partial v(k_i, \boldsymbol{\theta})}{\partial k_{jam}})^T$. The FIM is inversely correlated with the variance of the calibration parameters. It is a natural choice to design points where a large amount of information exists. Thus, the second criterion involves maximizing the determinant of $\mathbf{I}(\boldsymbol{k}, \boldsymbol{\theta})$ for a given $\boldsymbol{\theta}$, which has been proven to be the approximate locally D-optimal design [24]. The optimal FIM criterion is:

$$\boldsymbol{k}^\star = \underset{\boldsymbol{k} \in \mathbb{K}^N}{\operatorname{argmax}} |\mathbf{I}(\boldsymbol{k}, \boldsymbol{\theta}^\star)|, \tag{9}$$

where $|\mathbf{A}|$ is the determinant of matrix $\mathbf{A}$. By considering the above two objectives and aiming to obtain robust and accurate estimates, the design criterion becomes:

$$k^{\star} = \underset{k \in \mathbb{K}^N}{\operatorname{argmax}}\{\|\hat{\zeta}(k) - v(k, \theta^{\star})\| + \lambda|\mathbf{I}(k, \theta^{\star})|\}, \tag{10}$$

where $\lambda > 0$ is a hyperparameter, selected as described in Section 2.3.

*2.3. Algorithm for Generating a Sequential Optimal Sub-Design*

Since $\theta^{\star}$ is unknown in (10), $k$ and $\theta$ must be optimized simultaneously, with the most common solution being updating $k$ and $\theta$ iteratively using sequential methods. First, assuming that $\mathcal{D}_0 = \{k_1, \cdots, k_{n_0}\}$ is the initial design selected using the space-filling methods, and that the current density set is $\mathcal{D}_i = \{k_1, \cdots, k_i\}$, $\theta$ is estimated through optimizing the empirical $L_2$ loss function according to $\mathcal{D}_i$:

$$\hat{\theta}_i = \underset{\theta \in \Theta}{\operatorname{argmin}} L_f(\mathcal{D}_i, \theta), \tag{11}$$

where $L_f(\mathcal{D}_i, \theta) = \frac{1}{i}\sum_{r=1}^{i}(\hat{\zeta}(k_r) - v(k_r, \theta))^2$ presents the empirical $L_2$ loss function. Additionally, by fixing $\theta^{\star}$ and $\mathcal{D}_i$ in (10) to $\hat{\theta}_i$ and optimizing it, we obtain $k_{i+1} = \underset{k \in \mathbb{K}}{\operatorname{argmax}}\{|\hat{\zeta}(k) - v(k, \hat{\theta}_i)| + \lambda|\mathbf{I}(k, \hat{\theta}_i q)|\}$. It is widely believed that the design points should be evenly spread in the experimental space to achieve a comprehensive exploration. Thus, we use the grid search method to find the optimal sub-design, which avoids requiring many design points in the neighborhood, with space-filling designs being typically used in grid search methods to generate lattice points that are robust to the modeling choices. In this article, we illustrate the method using the maximin Latin hypercube design (maximin LHD) [20,26], but we will maintain this flexibility of choice for the experimenter. Let the candidate points generated using the maximin LHD be $\mathbb{K}^c = \{k_1^c, \cdots, k_M^c\}$; the optimal density is generated by the following equation according to the sequential criterion:

$$k_{i+1} = \underset{k \in \mathbb{K}^c}{\operatorname{argmax}}\{|\hat{\zeta}(k) - v(k, \hat{\theta}_i)| + \lambda|\mathbf{I}(k, \hat{\theta}_i)|\}. \tag{12}$$

This article uses the grid search method to select $\lambda$ dynamically. Assuming that the initial alternative points of $\lambda$ are $\lambda_1, \cdots, \lambda_t$, the hyperparameter alternatives are input into (12) to obtain $t$ optimal sub-designs, respectively. Let the optimal sub-designs obtained for the $i$th under different hyperparameters be $\mathcal{D}_{i1}, \cdots, \mathcal{D}_{it}$. Applying them to the $L_f$ loss, the objective hyperparameter is selected by minimizing:

$$\lambda_i^{\star} = \underset{j \in \{1, \cdots, t\}}{\operatorname{argmin}} L_f(\mathcal{D}_{ij}, \hat{\theta}_i). \tag{13}$$

where $L_f(\mathcal{D}_{ij}, \hat{\theta}_i) = \frac{1}{i}\sum_{r=1}^{i}(\hat{\zeta}(k_{rj}) - y^s(k_{rj}, \hat{\theta}_i))^2$. Since $\lambda$ is reselected at each sequential design, we call it a dynamic hyperparameter selection.

Finally, Algorithm 1 summarizes the proposed method for generating the optimal sub-design.

---

**Algorithm 1** Generating the sequential optimal sub-design for the $L_2$ calibration.

---

**Input:** Initial design $\mathcal{D}_0 = (k_1, \cdots, k_{n_0})$, traffic flow data $\mathbf{K} = (k_1, \cdots, k_n)^T$ and $\mathbf{V} = (v_1^t, \cdots, v_n^t)^T$, candidate design set $\mathbb{K}^c$, alternative hyperparameter $\{\lambda_1, \cdots, \lambda_t\}$, number of sequential additional points $m$.

**Initialize:** $\hat{\zeta}(\cdot)$ is given based on $\mathbf{K}$ and $\mathbf{V}$.

   **for** $i = 1$ to $m$ **do**

      **for** $j = 1$ to $t$ **do**

         $k_{ij} \leftarrow \underset{k \in \mathbb{K}^c}{\mathrm{argmax}}\{|\hat{\zeta}(k) - v(k, \hat{\boldsymbol{\theta}}_{i-1})| + \lambda_j|\mathbf{I}(k, \hat{\boldsymbol{\theta}}_{i-1})|\},$

         $\mathcal{D}_{ij} \leftarrow \mathcal{D}_{i-1} \cup k_{ij}.$

      **end for**

      $\lambda_{il}^\star = \underset{l \in \{1, \cdots, t\}}{\mathrm{argmin}}\, L_f(\mathcal{D}_{il}, \hat{\boldsymbol{\theta}}_j),$

      $k_i^\star \leftarrow k_{il},$

      $\mathbb{K}^c \leftarrow \mathbb{K}^c - \{k_i^\star\},$

      $\mathcal{D}_i \leftarrow \mathcal{D}_{i-1} \cup k_i^\star,$

      $\hat{\boldsymbol{\theta}}_i = \underset{\boldsymbol{\theta} \in \Theta}{\mathrm{argmin}}\, L_f(\mathcal{D}_i, \boldsymbol{\theta}).$

   **end for**

**Output:** Optimal sub-design $\mathcal{D}_m$ and calibration parameter estimate $\hat{\boldsymbol{\theta}}_m$.

---

### 2.4. Uncertainty Quantification of the Calibration Parameters

In practice, we aim not only to obtain the point estimates of the calibration parameters, but to gain the variance of the parameter estimates to quantify uncertainty. Since the dual-regime modified Greenshields model is deterministic, the model's uncertainty originates from $\hat{\boldsymbol{\theta}}$. Considering the frequency methods, the bootstrap methods have been widely used to calculate the variance and confidence intervals of the parameters [27]. The initial design is repeated for $T$ times, and the estimates of the calibration parameters are obtained using the proposed method. The specific steps are presented below:

- **Step 1:** $\mathbf{K}' = (k_1', \cdots, k_n')^T$ and the corresponding $\mathbf{V}' = (v_1^{t'}, \cdots, v_n^{t'})^T$ can be obtained using the replacement sampling method from the real traffic flow data $\mathbf{K}$ and $\mathbf{V}$.
- **Step 2:** The surrogate model $\hat{\zeta}'(\cdot)$ of the traffic flow system is estimated according to $\mathbf{K}'$ and $\mathbf{V}'$.
- **Step 3:** To estimate the calibration parameters according to Algorithm 1.
- **Step 4:** Repeat the above steps $T$ times to obtain $\{\hat{\boldsymbol{\theta}}_1, \cdots, \hat{\boldsymbol{\theta}}_T\}$, and compute their variance and empirical confidence interval.

## 3. Case Study

This section investigates the performance of the proposed method (abbreviated as Fast-$L_2$ calibration) on the traffic flow system of the M25 motorway in London. The London Orbital motorway is a circular highway around London, and since the M25 is the busiest motorway in the UK and traffic jam is relatively severe [28], we select it for our study. Section 3.1 present the sources and description of the traffic flow data, Section 3.2 introduces the settings of various calibration methods, and Section 3.3 presents the corresponding calibration results.

### 3.1. Data Source of the Traffic Flow Model

The primary source of traffic data is obtained through loop detectors installed in the highway lanes, and such data are available from several web-based data archiving systems. This work utilizes real and simulated data downloaded from http://tris.highwaysengland.co.uk/detail/trafficflowdata (accessed on 24 September 2022), which contains historical traffic data at 15 min aggregation intervals on the M25 motorway in London from 1 to 5 June 2021. Figure 2 illustrates the distribution of the selected loop detector locations in the study area, and Figure 3 depicts the scatterplot of the record traffic flow data. Figure 3 highlights that when the density is relatively small, the vehicle's speed remains around

110 km/h. As the density increases, traffic jams occur, and the speed gradually decreases to the minimum value.

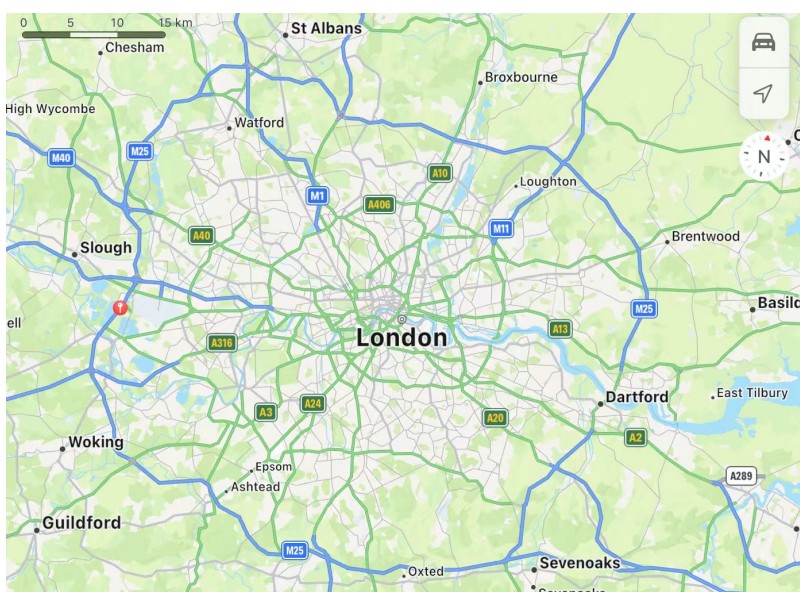

**Figure 2.** Maps of the selected detector locations on the M25 motorway in London, where the big red pin indicates the detector.

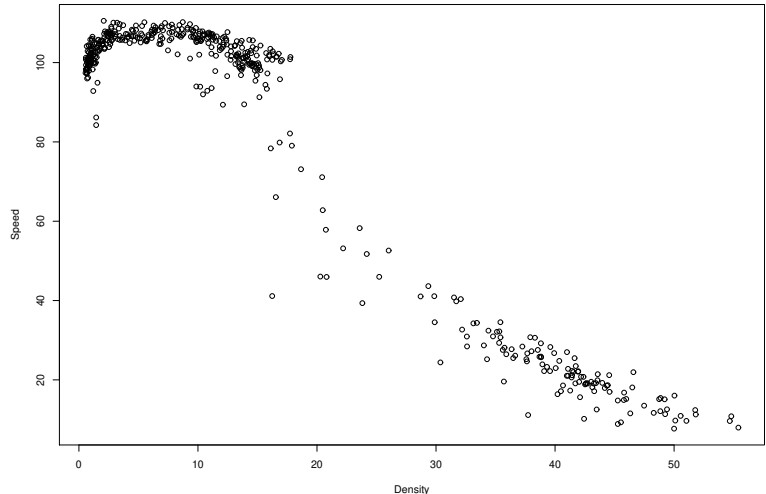

**Figure 3.** Speed vs. density scatterplot from 1 to 5 June 2021.

In [6,29], the authors used the modified Greenshields model to calibrate highways in the United States. Based on their experience and scatter plots, the value regions for the six calibration parameters are reported in Table 1.

**Table 1.** Value regions for the calibration parameters.

| $\theta$ | $k_{bp}$ | $u_f$ | $v_f$ | $\alpha$ | $v_0$ | $k_{jam}$ |
|---|---|---|---|---|---|---|
| Value region | [10, 30] | [80, 130] | [170, 220] | [0, 10] | [0, 5] | [200, 220] |

### 3.2. The Settings of the Calibration Methods

To verify our proposed method's performance, we challenge it against the $L_2$, KO [9], and the LS [11] calibration methods. However, due to the computational burden of the $L_2$ calibration, in the comparisons, we use the projected $L_2$ calibration (Proj-$L_2$) [30] variant,

which is the $L_2$ calibration method under a Bayesian framework. Given that the true calibration parameter values cannot be calculated in this case study, to evaluate the performance of the calibration methods, we first use the relative prediction discrepancy (RPD) as the statistical criterion to compare the different approaches. The RPD determines the prediction accuracy for the calibrated computer model, defined as follows:

$$\text{RPD} = \frac{1}{M} \sum_{i=1}^{M} \left\{ \frac{1}{n_{test}} \sum_{j=1}^{n_{test}} \left| \frac{v_j^t - v(k_j, \hat{\boldsymbol{\theta}}_{ij})}{v_j^t} \right| \right\}, \tag{14}$$

where $M = 50$ is the repetition and $|\cdot|$ is the absolute value. $\{k_1, \cdots, k_{n_{test}}\}$ and $\{v_1^t, \cdots, v_{n_{test}}^t\}$ are the testing sets, with $n_{test}$ being the sample size.

The initial design is fixed at the same sample size, which is changed at each replication to calculate the RPD. For the proposed method, the initial design size is set to $n_0 = 2q$ and is obtained using the maximin LHD method from $\mathbb{K}$, where $q$ is the dimension of the calibration parameters. The number of additional sequential points is $m = 5q$ obtained on Algorithm 1. For a fair comparison, the sample size is set as $N = 7q$ for the KO, LS, and Proj-$L_2$ calibration methods, which is the same as the total sample size after adding points for the proposed method. $n_{test}$ is set on $n_0$ and the testing data are selected from the real traffic flows data randomly. For the Fast-$L_2$ calibration, we use the scaled Gaussian process [21] to estimate the real traffic flow system, and the **RobustGasp** package [31] in R is employed to build the scaled Gaussian Process model. The variance and running time are also used as guidelines for comparing the performances of different calibration methods. Since the Fast-$L_2$ and LS are frequency methods, we use 500 bootstrap samples to measure their variances and running times. For the KO and Proj-$L_2$ calibrations, the prior density of $\theta$ is set as an uninformative prior. Additionally, the $r(\cdot, \cdot)$ in the benchmark methods is set on the Matérn kernel function with a smooth parameter $\nu = 5/2$, and the scaling parameter $\psi$ is fixed to $1/2$. The variances of KO and Proj-$L_2$ are calculated using posterior samples of the calibration parameters.

### 3.3. The Results

Table 2 reports the RPD, the mean standard deviation (mSD) of $\hat{\theta}$, and the runtime, which are used to compare the prediction discrepancy and computational efficiency of the four methods.

**Table 2.** Summary statistics of $\hat{\theta}$ for different calibration methods.

| Calibration Methods | Fast-$L_2$ | KO | LS | Proj-$L_2$ |
|---|---|---|---|---|
| RPD | 0.7039 | 2.6187 | 0.8345 | 6.7197 |
| mSD | 1.3377 | 3.0554 | 2.1423 | 6.8666 |
| Runtime | 14.28 s | 40.67 s | 0.22 s | 637.55 s |

Due to the ideal assumptions and simplifications, the modified Greenshields model is inexact; that is, the RPDs of four different calibration methods are relatively large. Among the four benchmark methods, the RPD of the Fast-$L_2$ calibration method is the smallest, affording the best prediction accuracy. According to (14), the RPD of LS is theoretically smaller than Fast-$L_2$. Since the Fast-$L_2$ chooses more efficient sample points according to the proposed optimal criteria, its RPD is smaller than the LS. The RPD of KO's calibration is larger than Fast-$L_2$ and LS because the KO does not converge to the true value when the discrepancy function exists [17]. The RPD and mSD of the Proj-$L_2$'s calibration are the largest due to the inaccurate Gaussian process estimation. Moreover, Fast-$L_2$ has the smallest mSD, indicating that it provides the smallest $\hat{\theta}$ uncertainty. Finally, LS requires the shortest time due to simple calculations, and the runtime of Fast-$L_2$ only requires 14.28 s, which is much smaller than Proj-$L_2$ calibration time.

To further compare the uncertainty of each calibration parameter, Figure 4 illustrates the box plots of $\hat{\theta}$ using different calibration methods. Combined with the mSD in Table 2, it highlights that the proposed calibration parameter estimation has the smallest variance. That is, the uncertainty provided by the Fast-$L_2$ calibration is smaller than the competitor calibration methods. The estimated value of the Fast-$L_2$ is close to the LS, and although the true value of $\theta$ is unknown, the estimates of Fast-$L_2$ and LS are more accurate than the other RPD-based methods and the $\theta$ estimates.

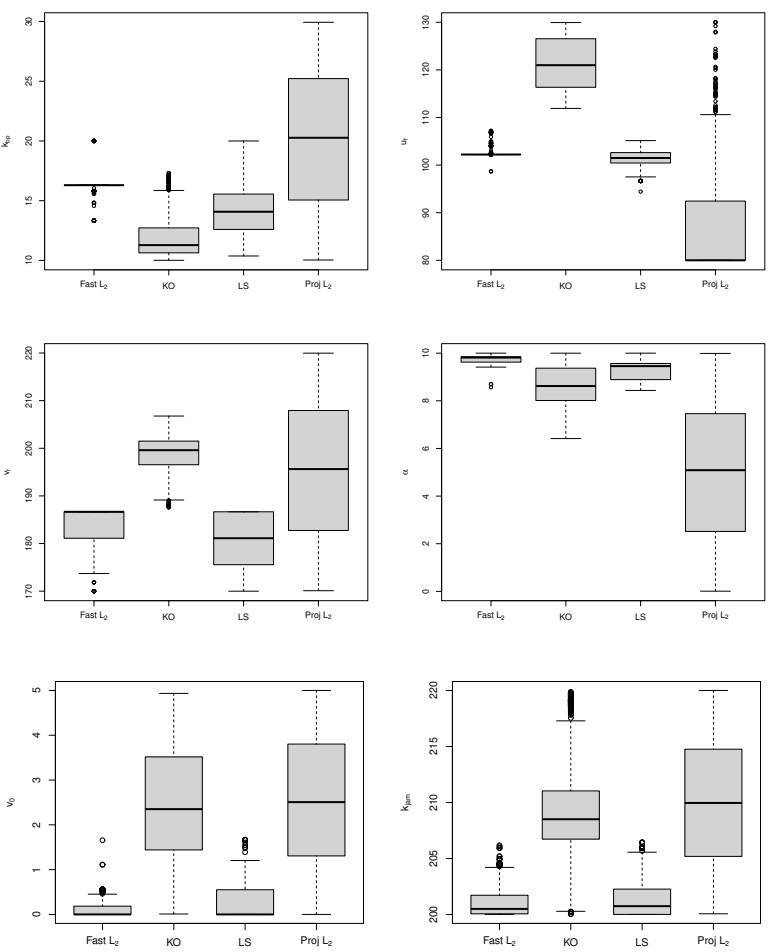

**Figure 4.** Box plots for four calibration methods in case study.

Figure 5 depicts the predictions and confidence interval of the modified Greenshields model after calibration, according to the optimal sub-design criterion. The results infer that the computer model fits the observations well and that the 95% interval is narrow. Additionally, the scatter plots of the testing data and the predicted values are uniformly distributed around $y = x$, with the coefficient of determination $R^2 = 0.98$.

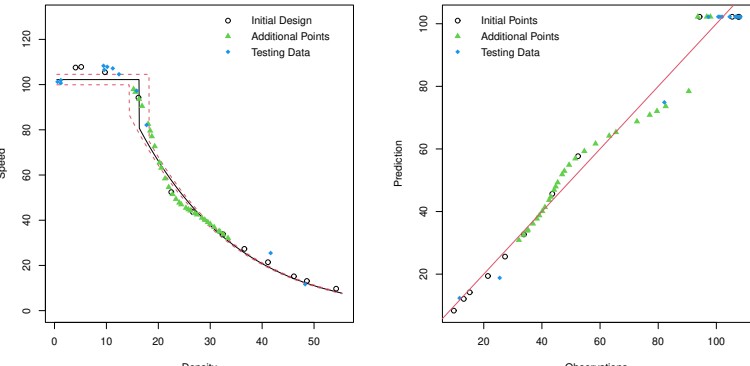

**Figure 5.** (**Left**): Computer model after calibration and the 95% prediction confidence interval, using the proposed method. (**Right**): Observations vs. predictions of the calibrated computer model.

## 4. Conclusions

This work proposes a fast $L_2$ calibration framework suitable for the inexact traffic flow system. The proposed method first suggests an optimal sub-design criterion for the $L_2$ calibration based on the discrepancy function and FIM, which reduces the computational time and preserves the advantages of $L_2$ calibration. Considering the space-filling of the design, we employ the grid search method to find the additional points sequentially. Then, we develop an algorithm to generate the optimal design, and a standard bootstrap method is utilized to quantify the uncertainty of the predictors. Finally, we apply the proposed method to a case study of the M25 motorway in London. The results demonstrate that the prediction accuracy of the calibration parameters estimated based on our optimal design criterion is better than that of the current calibration methods. Furthermore, the suggested method significantly improves the computational efficiency of the $L_2$ calibration and reduces the calibration parameters' uncertainty. The results demonstrate that the proposed method applies to inexact traffic flow models.

The future research directions are multifaceted. First, since most data have periodicity, which is not considered in our paper, an optimal design criterion for periodic data can be developed in the future. Second, an optimal design criterion under the Bayesian version can be considered as being more convenient for quantifying the uncertainty.

**Author Contributions:** Conceptualization, Y.W.; writing—original draft preparation, J.H.; writing—review and editing, J.H., Y.W. and M.H.; funding acquisition, Y.W. All authors have read and agreed to the published version of the manuscript.

**Funding:** Wang's research was supported by the Natural Science Foundation of Beijing Municipality (1214019).

**Institutional Review Board Statement:** Not applicable.

**Informed Consent Statement:** Not applicable.

**Data Availability Statement:** Not applicable.

**Conflicts of Interest:** The authors declare no conflict of interest.

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
