# Peer review of "Fast L2 Calibration for Inexact Highway Traffic Flow Systems"

_electronics, doi:10.3390/electronics11223710_

Round 1
Reviewer 1 Report
This paper proposes a fast L 2 calibration framework for an inexact traffic flow model, Greenshields. The topic is interesting, and the paper is well-written. I have some questions about the paper, please see the following comments.
1. What are x_i and y_i in equation 2?
2. What is k in equation 4?
3. Page 7, line 206, typo: “table 2”.
4. Is the speed in figure 3 the average of the whole loop of M25?
5. From the results shown in figure 2, LS seems to be very efficient, which could make it real-time applicable (maybe to solve the periodicity problem). Is there any way that L2’s speed can be further improved?
Author Response
Dear Editor:
Please see the attachment.
Special thanks to you for your good comments!

Reviewer 2 Report
This is an interesting study to develop a calibration framework for the inexact traffic flow system. With this approach the computational time reduces with accurate results. Below are some of my suggestions to improve the paper:
1. Ln 53: needs more background and literature about the L2 calibration method by adding other studies referring the benefits of the method.
2. Ln 189: the busiest motorway in the UK sentence should be supported with numbers or references
3. Ln 199: Figure 2 does not have the locations of loop detectors
4. Ln 201: Not both figures highlight the density and vehicle speed correlation, it is only Figure 3.
5. Ln 201: what authors meant by physical observations? did they collect data in the field?
6. Figure 2: needs to add loop detectors, scale, north arrow and with the loop detectors a legend.
7. Ln 206: needs to be Table 1
8. the comparison of the models need to be expanded. It would be better to show how accurate the findings of the new approach.
Author Response

(The authors gave the same response as above.)
